**Data Availability Statement:** Data has been made available at https://github.com/nicholashorne1/crm/tree/Mortality-Estimates/data.

**Funding:** This research was funded under the Bryden Centre. https://www.brydencentre.com/.

# Providing a detailed estimate of mortality using a simulation-based collision risk model

**Nicholas Horne**[ID]¹*, **Ross M. Culloch**²,³, **Pál Schmitt**⁴, **Ben Wilson**⁵, **Andrew C. Dale**⁵, **Jonathan D. R. Houghton**⁶, **Louise T. Kregting**¹

1 School of Natural and Built Environment, Queen's Marine Laboratory, Queen's University Belfast, Newtownards, Northern Ireland, United Kingdom, 2 Marine Scotland Science, Scottish Government, Marine Laboratory, Aberdeen, Scotland, United Kingdom, 3 APEM Limited, Heaton Mersey, Stockport, United Kingdom, 4 School of Chemistry and Chemical Engineering, Queen's Marine Laboratory, Queen's University Belfast, Newtownards, Northern Ireland, United Kingdom, 5 Scottish Association for Marine Science (SAMS), University of the Highlands and Islands, Oban, Argyll, Scotland, United Kingdom, 6 School of Biological Sciences, Queen's University Belfast, Belfast, Northern Ireland, United Kingdom

* nhorne01@qub.ac.uk

## Abstract

Marine renewables could form a significant part of the green energy mix. However, a potential environmental impact of tidal energy converters (TECs) is collision risk between a device and animal, which has been a significant barrier in the consenting process. While it is important to understand the number of collisions of an animal with the device, the relative speed at which an animal collides with the device, and the point on the animal where collision occurs, will determine whether a collision is fatal, which is important in understanding population-level impacts. Using a simulation-based collision risk model, this paper demonstrates a novel method for producing estimates of mortality. Extracting both the speed and the location of collisions between an animal and TEC, in this instance a seal and horizontal axis turbine, collision speed and location of collision are used to produce probabilities of mortality. To provide a hypothetical example we quantified the speed and position at which a collision occurs to estimate mortality and, using collision position, we determine all predicted collisions with the head of the animal as fatal, for example, whilst deeming other collisions non-fatal. This is the first collision risk model to incorporate speed at the point of contact and the location where the collision occurs on the animal, to estimate the probability of mortality resulting from a collision. The hypothetical scenarios outline how these important variables extracted from the model can be used to predict the proportion of fatal events. This model enables a comprehensive approach that ultimately provides advancements in collision risk modelling for use in the consenting process of TECs. Furthermore, these methods can easily be adapted to other renewable energy devices and receptors, such as wind and birds.

## Introduction

As countries strive to meet climate preservation targets, such as those outlined in the Paris Agreement [1], this has favoured the expansion of the renewable energy industry across several

The funders had no role in study design, data collection and analysis, decision to publish, or preparation of the manuscript.

**Competing interests:** The authors have declared that no competing interests exist.

regions of the world. Renewable energy production has risen by 440% between 2009 and 2019 [2] with the wind and solar industries making the largest contribution [2]. However, there are untapped marine energy resources, such as tides and waves, that could also be utilised as part of the energy network [3]. The predictable nature of the tides offers a reliable supply of renewable energy, making tidal energy converters (TECs) a desirable source of electrical generation which can provide a baseload supply. There are a range of tidal turbine designs for extracting energy from the marine environment, such as horizontal axis turbines [4], crossflow turbines [5] and tidal kites [6], targeting regions with high flow speeds (ca. 2-4ms$^{-1}$). However, these high flow environments can also be important habitats for many protected marine species [7], such as harbour porpoise (*Phocoena phocoena*) [8], orca (*Orcinus orca*) [9], harbour seal (*Phoca vitulina)* [10] and sockeye salmon (*Oncorhynchus nerka*) [11]. Therefore, robust environmental impact assessments (EIAs) are typically required by regulators to quantify potential impacts of TECs on protected populations and habitats.

A key issue when considering potential environmental impacts of TECs is collision risk between the device(s) and animals; this has, and continues to be, a potentially significant barrier in consenting these devices [12]. Estimates of collision risk are calculated using collision risk models (CRMs) which are used to predict the potential risk posed to animals at a population or management unit level [13]. However, there is still a level of uncertainty in collision risk estimates, largely due to the difficulty in validating models by directly witnessing any potential collisions and the often limited information and data available on fine-scale behaviour of animals around devices [14–16]. CRMs currently make use of empirical data such as those pertaining to device characteristics and animal size, to produce probabilities of collision [13]. Broadly, the estimates produced by the CRMs currently employed calculate the probability of a single transit by an animal leading to a collision which is then scaled up to a population level or management unit estimate of impact. Also, multiple values (e.g. 50%, 75%, 90%, 98%) are often applied to these CRM estimates for the assumed evasion and avoidance by the animals with a range of numbers chosen to address the large level of uncertainty around these behavioural parameters [17].

While it is recognised that monitoring fine-scale animal behaviour around TECs is key to understanding the number, speed, and location of collisions (on the device and animal), it is the relative speeds of animal and device, and the point on the animal's body where collision occurs, that will determine which collisions are likely to be fatal. To our knowledge, there are only three published papers that have investigated consequences of a collision with TECs, all of which used animals *post-mortem*, recovered from strandings [18–20]. Carlson *et al* [18] and Copping *et al* [19] showed, using tensile testing of tissue samples, that for orca, collisions were unlikely to be fatal irrespective of where the strike occurred on the animal's body [18] and for harbour seals, demonstrated a range of scenarios where mortality was likely, such as collisions with a tip speed over 6.5ms$^{-1}$ [19]. However, for both Carlson *et al.* [18] and Copping *et al.* [19] the physiology was modelled by simulating tissue structure to estimate the likelihood of severe injury and therefore did not consider any injuries that may occur to the organs or bone structure of the animal.

Using harbour and grey seal (*Halichoerus grypus*) carcasses and striking them at the sea surface with a tidal turbine blade attached to a small vessel, Onoufriou *et al.* [20] investigated at which relative speed and where on the seal's body severe trauma would likely occur based on pathological examination of the carcasses post-collision. They predicted that pathological effects likely to cause mortality occurred at speeds greater than 5.1ms$^{-1}$ (95% C.I. = 3.2–6.6 ms$^{-1}$). The results from these experiments were then used to refine a collision risk assessment by calculating the percentage of time the blade spent over 5.1 ms$^{-1}$. However, those calculations were based on 2D modelling that did not consider the 3D shape of the animal and the relative speed

that occurs from two moving objects colliding, therefore potentially underestimating the likelihood of mortality. Furthermore, the issue of concussion was unable to be investigated by Onoufriou *et al* [20], which is important, as there is the potential for lower collision speeds to cause concussion, resulting in marine mammals drowning, which would inflate the number of fatal collisions. Consequently, having a collision risk model with the ability to extract the location of a strike on the body of the animal, as well as the relative speed of collision, would be a valuable tool to better predict the likelihood and consequences of collisions.

To date, the incorporation of the speed and point of collision of an animal with a device has not been possible in previous CRMs [13, 20]. Here we provide a step change in the development of the simulation-based approach [21, 22] to offer a method for estimating the probability of mortality in collision risk assessments. The approach uses an open-source 3D modelling and game-design software called Blender [23]. The software has a collision detection system which enables additional information to be gathered when a collision occurs, such as the speed and location of a collision on both the device and animal. In doing this, outputs can be used to determine the likelihood of mortality occurring and so produce a distribution of data from which the risk of mortality can be assessed.

This paper demonstrates how the simulation-based approach can be used to incorporate mortality estimates into a collision risk assessment. Herein, a novel method is outlined for a) extracting collision speed between a horizontal axis turbine and a seal, b) extracting the location of collision on the animal and c) a hypothetical example is given for applying the information on collision speed and location of collision to produce probabilities of mortality.

## Materials and methods

### Simulations

Blender, an open-source 3D modelling and game-design software, was interfaced using a Python [24] script to setup and run simulations. The movements of an animal and TEC were simulated in 3D space, which was repeated over many simulations to calculate a collision probability (CP) over the swept area of the TEC. The simulations presented herein are of a seal moving horizontally downstream towards a rotating three-bladed horizontal-axis tidal turbine (HATT). The seal starting positions were set to ensure a uniform distribution of 0.5m intervals over a 22m by 23m area that covered the full swept area of the HATT, (i.e., the size of the rotor blade, 18m, and the foundation of the device). The TEC was based on the dimensions and characteristics of the Siemens Atlantis AR1500 HATT and was simulated using the same 3D shapefile for the 18m diameter rotor blade used in Horne *et al* [21]. The seal was based on the dimensions of an adult harbour seal (Length = 1.41m, Width = 0.34m) [13]. For full details on the formulation and accuracy of the model see Horne et al. [21, 22].

The effect of two factors on the collision speed were investigated: the approach speed of the animal and the rotational speed, in rotations per minute (RPM), of the device rotor. Three biologically feasible approach speeds, equivalent to the animal's speed over ground were used. Low speed ($0.5 \text{ms}^{-1}$) to represent an animal moving against the tidal current, a behaviour which has been recorded in Strangford Lough [25]. Mid speed ($1.8 \text{ms}^{-1}$) based on the Scottish Natural Heritage guidance for the mean swim speed of a harbour seal to be used in a CRM [13]. High speed ($4.0 \text{ms}^{-1}$) where a seal is moving with the tidal flow [25]. Two rotational speeds were chosen to represent the cut-in speed, at a tidal flow of approximately $1.0 \text{ms}^{-1}$, where the HATT begins rotating and reaches speeds of 8RPM, and the operational speed of around $3.4 \text{ms}^{-1}$, rotating at approximately 14RPM [17]. The RPMs associated with the cut-in and operational speeds were used to calculate the rotations in radians per second (RPS); the

unit used to set the rotational speed of objects within Blender, which is calculated as:

$$RPS = \frac{RPM \times 2\pi}{60} \qquad \text{(Eq 1)}$$

Where the number of times the blade completes a 360˚ rotation ($2\pi$) in one minute (*RPM*) divided by the number of seconds in a minute (60) produces the radians per second (*RPS*) (Eq 1). We modelled 211,500 simulations; these consisted of 21,150 starting positions with 100 time-lags per starting position. Time-lags, detailed further in Horne et al. [21], are used to calculate a probability for each starting position and, in undertaking a convergence study [22], 100 time-lags were deemed sufficient for the scenarios tested here.

A Python script was developed to extract the collision speed, point of contact on the animal and the location of the centre of the animal shape for each collision that occurs during the simulations. Point of collision on the animal was calculated by subtracting the location of the centre of the animal shape from the point of contact. The collisions speed was calculated within the Python script by taking the magnitude after the addition of the two velocities of the seal (*VSeal*) and device (*VDevice*) at the point of collision as seen in Eq 2.

$$Collision\ Speed = Magnitude(VSeal + VDevice) \qquad \text{(Eq 2)}$$

These results, in addition to the original input parameters for each individual simulation, were output to a CSV file for further analysis. Each line of the CSV file corresponded to an individual collision and contained the collision speed, the point of contact on the TEC and animal, and an individual ID number for the simulation that corresponded to the input parameters for that individual simulation. The ID number of the simulations were used to match the results to the input file to generate a file that contained all the information for each scenario.

## Analysis

**Collision speed.**   Using R [26] the collision speed was explored to demonstrate the difference in collision probabilities between the cut-in speed scenarios and the operational speed scenarios. Collision speed over the device was also investigated by plotting the distribution of collision speeds at the points of collision. These plots were produced using ggplot2 [27].

**Collision position.**   The point of contact on the animal was classified as either 'head' or 'torso'. From head to tail, a point 0.4 m behind the nose of the animal was chosen as the difference between collision with the head (<0.4 m) and torso (>0.4 m). The distinction between head and torso were then used to calculate probabilities with different levels of precaution considered. For example, if a collision occurs on the head of an animal, this could be taken as indicative of a higher probability of mortality i.e. through increase probability of concussion and subsequent drowning. In this study one precautionary scenario considered that all head collisions are fatal, irrespective of the speed of collision.

**Mortality thresholds.**   To investigate how the speed of collision affects collision risk, a variety of hypothetical thresholds were applied to calculate mortality probabilities. For example, a collision speed threshold could be set to 4 ms$^{-1}$ meaning any collision over that speed would be classified as fatal. Multiple hypothetical collision speed thresholds (from 0 to 7 ms$^{-1}$) were applied to all scenarios to investigate how different thresholds influenced mortality probabilities.

In addition to applying a collision speed threshold, the position on the animal that a collision occurs was incorporated into the mortality probability estimate by classifying all head collisions as fatal and applying collision speed thresholds only to collisions with the torso. Graphical outputs were produced using ggplot2 [27] across all the scenarios tested.

## Results

### Collision speed

For each of the rotational speeds (RPMs) and seal approach speeds, distributions of collision speeds were obtained (Fig 1). The lowest collision speed for each scenario occurred at the hub (centre) of the device and increased linearly towards the tip of the blades (Fig 1; black arrows). When comparing distributions, the cut-in scenarios (Fig 1; left) showed slower collision speeds, as compared to the operational speed scenarios, which had collision speeds as high as 17 ms$^{-1}$ (Fig 1; right).

The distribution of collision speeds varied with scenario, where some had a more even spread of collision speeds (Fig 1F) and some had a more distinct peak (Fig 1A). An even spread represents collisions occurring evenly across the device rotor, whilst the scenarios that have peaks in their distribution have collisions occurring more often either near the hub of the device (Fig 1C) or near the fast-moving tips of the blades (Fig 1D).

### Point of contact

The point of contact is the position on the seal where the collision occurred, and this was assessed for accuracy by plotting the distribution from a side on view of the seal (Fig 2). The collision positions on the X-axis, which represents the head to tail dimension of the seal, were deemed to be within reasonable bounds due to a low error rate (error < 0.09%). The error rate was calculated from the percentage of collisions with a point of contact beyond the front of the seal.

Over the range of scenarios, the number of collisions that occurred to the head of the animal compared to the torso changed dependent on both the approach (A) and rotational speed (R) (Table 1). A faster moving animal resulted in fewer collisions, whilst for most cases a faster device RPM resulted in more collisions. The mid approach speed scenarios for both rotational speeds showed a lower percentage of head collisions when compared to the low and high approach speeds (Table 1). This indicates a non-linear pattern between animal speed, and the percentage of collisions with the head or torso. For example, when considering the operational speed (R2, Table 1), the low-speed scenario (A1) had 53.0% of collisions with the head which dropped to 26.1% for the mid approach speed scenario (A2). However, this trend did not extend through to the high-speed scenario (A3), where the percentage of head collision rose to 36.3%, indicating a more complex relationship between animal speed, RPM, location of collisions and the number of collisions. Collisions with the head (area below the black line) that occur (Fig 1) make up a lower proportion of collisions the higher the speed (i.e., the further towards the blade tip) that collisions occur.

It should be noted that the total number of collisions (Table 1) are an important context to all results when looking at the percentage of head vs torso collisions. Furthermore, the total number of collisions for A1R1 are slightly greater than A1R2, which is counterintuitive due to the higher RPM for A1R2, however this is likely due to the minor error margin in results due to the number of time lags run in these simulations, for further information on error margins and time lags see Horne et al. [21].

### Mortality thresholds

Fig 3A shows a slow-moving animal (0.5 ms$^{-1}$) with no collision speed threshold applied, which produced a uniform collision probability close to 1.0 across the entire swept area. When thresholds of 4.0 ms$^{-1}$ (Fig 3B) and 5.1 ms$^{-1}$ (Fig 3C) were applied, the low-speed collisions at the hub were deemed non-fatal, leading to a reduced mortality probability. However, when

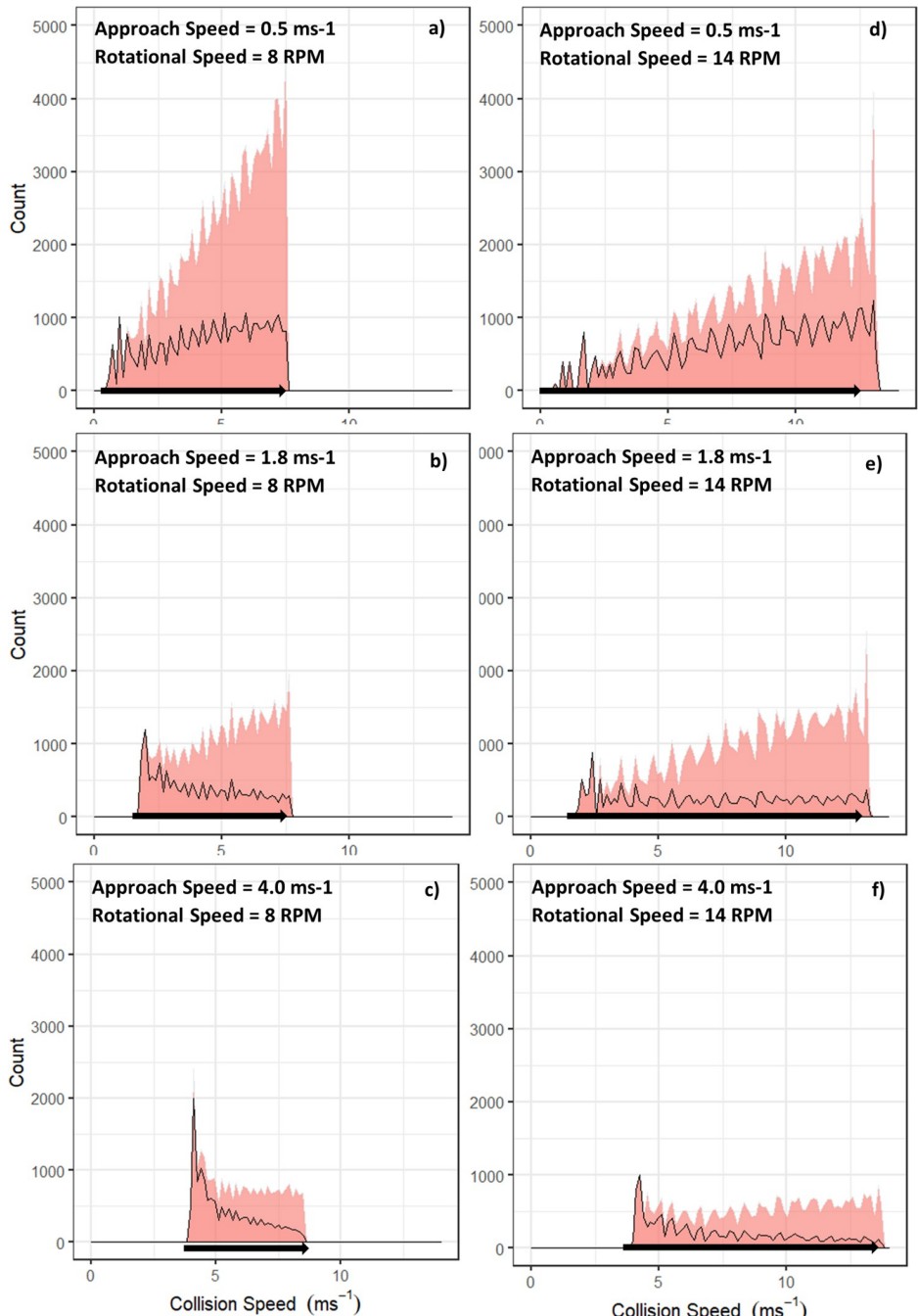

**Fig 1. Area plots displaying the total number of collisions (area) and the number of times (count) varying collision speeds (ms$^{-1}$) occurred.** The area under the black line are collisions with the head and the area above are collisions with the torso of the animal. Each graph represents a different scenario with approach and rotational speeds displayed on the corresponding graphs and the black arrows on the x-axis represent how the speed of collision increases from the hub to the tip of the blade (tip of the arrow).

collisions with the head were deemed fatal regardless of the collision speed and only collisions with the torso were deemed fatal for speeds over 5.1 ms$^{-1}$, this led to some collisions close to the hub being defined as fatal (Fig 3D).

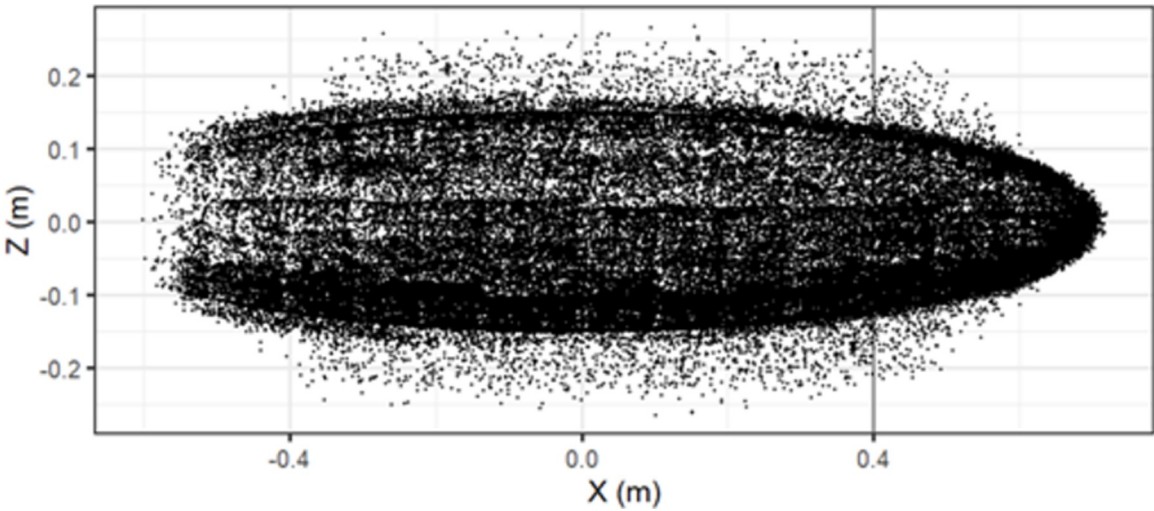

**Fig 2. Points of collisions on the seal from a side on view, the X-axis displays the length of the seal object from head to tail and the Z-axis is the virtual width of the seal from dorsal to ventral.** The threshold for head/torso collisions is displayed by the black line at 0.4m on the X-axis.

A range of thresholds from 0 ms$^{-1}$ to 7 ms$^{-1}$ were used to examine what effect a variety of collision speed thresholds might have on the different scenarios (Fig 4). The mortality probabilities in this regard were intuitive, as a less conservative threshold results in a reduced probability of mortality for each scenario. Furthermore, the change in scenario from a low approach speed (Fig 4A) to the mid approach speed (Fig 4B) had a large effect on mortality probability before any thresholds were applied, with a reduction in probability from 0.98 to 0.46. The way collision speed thresholds affect probabilities for different animal speeds can be demonstrated; for example, for a faster speed of approach (Fig 4B), mortality probabilities (if head collisions were deemed fatal) were not reduced by collision speed thresholds up to 2 ms$^{-1}$ due to the lowest collision speed being above 1.8 ms$^{-1}$ (i.e., the speed the animal was moving at); therefore, all collisions were deemed fatal.

As expected, when thresholds are applied to all collisions, and collisions with the head that occur below the threshold speed are regarded as fatal, there is a higher mortality probability. For example, for the A1R1 scenario this increased the mortality probability at a threshold of 5 ms$^{-1}$ from 0.57 to 0.75 (Fig 4A). Therefore, including all head collisions in the mortality probability calculation, irrespective of the collision speed threshold used, did provide a more conservative estimate. Furthermore, in these examples the conservative nature of this approach increases with collision speed, for example, for A1R1 at a threshold of 3 ms$^{-1}$ the head collisions would account for 8.8% of the mortality estimate whilst at a threshold of 7 ms$^{-1}$ they would account for 66.6% of the mortality estimate.

**Table 1. The total number of collisions in each scenario, and the % of those that were to the head or torso of the seal.** The scenarios refer to the codes, where A1R1 represents the scenario for the slowest approach speed (A1) and the cut-in rotational speed (8RPM) (R1).

| Scenario | Rotational Speed | RPM | Approach Speed | Speed | Head | Torso | Total |
|---|---|---|---|---|---|---|---|
| A1R1 | Cut-in | 8 | Low | 0.5 ms$^{-1}$ | 33.8% | 66.2% | 103,576 |
| A2R1 | Cut-in | 8 | Mid | 1.8 ms$^{-1}$ | 35.1% | 64.9% | 47,688 |
| A3R1 | Cut-in | 8 | High | 4.0 ms$^{-1}$ | 51.2% | 48.8% | 27,020 |
| A1R2 | Operational | 14 | Low | 0.5 ms$^{-1}$ | 53.0% | 47.0% | 102,459 |
| A2R2 | Operational | 14 | Mid | 1.8 ms$^{-1}$ | 26.1% | 73.9% | 77,773 |
| A3R2 | Operational | 14 | High | 4.0 ms$^{-1}$ | 36.3% | 63.7% | 38,835 |

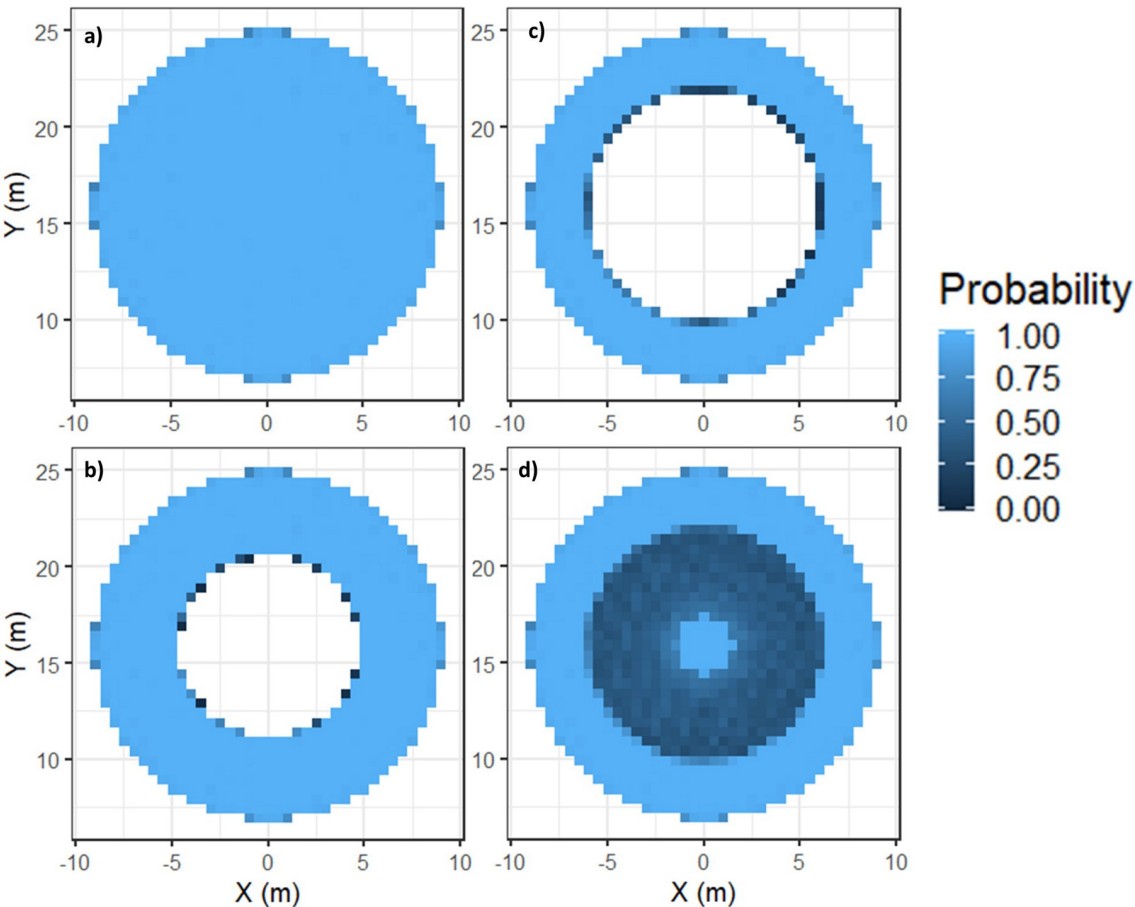

**Fig 3.** Probability of fatal collisions over the swept area of the device for scenario A1R1 for probabilities calculated with a) no collision speed threshold set (i.e., all collisions are fatal) b) above a collision speed threshold of 4.0 ms$^{-1}$ and c) above a collision speed threshold of 5.1 ms$^{-1}$ and d) all collisions with the head and collisions with the torso above a collision speed threshold of 5.1 ms$^{-1}$.

## Discussion

This work improves our ability to estimate whether mortality occurs when an animal collides with a TEC, which is an important step forward for collision risk assessments. Here we demonstrated how a novel simulation-based CRM can be used to extract speed at the point of collision and where the location of the collision occurs on the body of the animal and on the device. The ability to extract this information gives us a greater level of information for what we have now demonstrated are significantly important variables to consider when providing predictions of collision risk and mortality events. In the hypothetical example used herein, we demonstrated variation in mortality probabilities depending on collision speed thresholds and point of impact on the animal. This paper provides an example where any mortality thresholds may be applied due to the flexible nature of the simulation-based approach. This is an improvement on earlier CRMs which only produced a single value for collision risk. Furthermore, the simulation-based CRM also allows for the incorporation of other parameters including those relating to animal ecology and behaviour (e.g., dive patterns [22]) and different TEC designs (e.g., tidal kites [21]). Given the flexibility of this approach, this collision risk model can be applied to many scenarios and is not limited to a particular industry or receptor type.

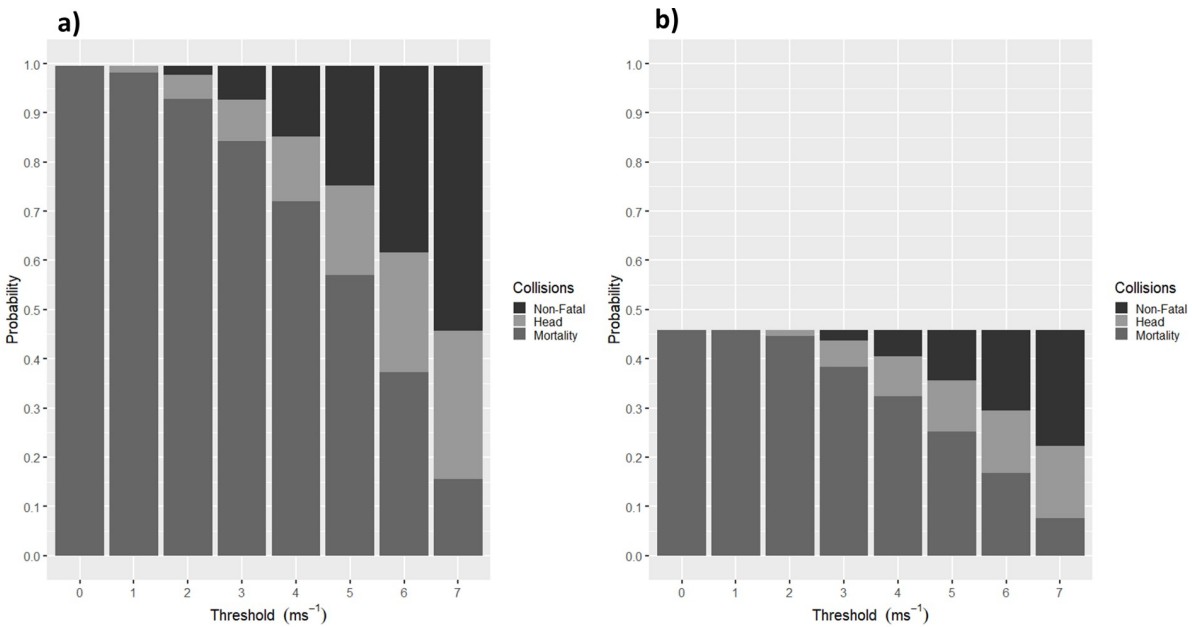

**Fig 4.** The probabilities for scenario a) A1R1 and b) A2R1 with a variety of collision speed thresholds applied from 0 to 7 ms$^{-1}$. The dark grey represents the mortality probability, which is classified as a collision that occurs at a greater speed than the threshold set, and the black represents the probability of an animal colliding at a speed lower than the threshold set and, therefore, is deemed to survive the collision. The light grey represents the probability that is made up of collisions above the threshold but where the collision was to the head of the animal.

Variations to animal speed and device RPM affected the collision speed and where on the body of the animal collisions occurred. An increase in approach speed of the animal did not necessarily result in higher relative collision speeds. This may seem counterintuitive, but an animal is less likely to collide near the tip of the blade (the fastest moving part of the device) when it is approaching at faster speeds. Conversely, when the blades of the device are rotating faster, it will increase both the collision speed and the probability of a collision occurring, relative to the speed in which the animal is travelling. This illustrates the fact that the relationship between mortality and collision speed is complex, as changes to parameters (e.g., approach speed) will affect both the speed of collisions but also the number of collisions occurring. Consequently, this is not a linear relationship, at least, in the scenarios tested herein. Whilst this paper showed these complex relationships, only two rotational speeds and three animal speeds were investigated where, in reality, devices and animal could move at any speeds between those investigated. Further investigation into more speeds may improve our understanding of the complex relationships.

Where the collision occurred on the body of the animal varied depending on the speed of the animal and the device. It may be expected that the point of contact on the animal would follow a simple rule, such as an increase in animal speed reduces the percentage of collisions with the head. However, compared to the low (0.5ms$^{-1}$) and high (4ms$^{-1}$) approach speeds, the mid approach speed (1.8ms$^{-1}$) for both rotational speeds showed the lowest percentage of collisions with the head. This non-linear relationship occurs because the point of collision on the animal will depend on the speed of the animal, the speed of the device and where on the device collisions occur. When the seal is moving slowly (0.5ms$^{-1}$), the chance of a head collision is increased as the time taken to pass by the rotating blade is longer (Fig 5A). Device speed is important in this regard, as a faster moving blade will be more likely to strike the head of a slow-moving animal as the animal cannot pass enough of the rotating blade in time. When the

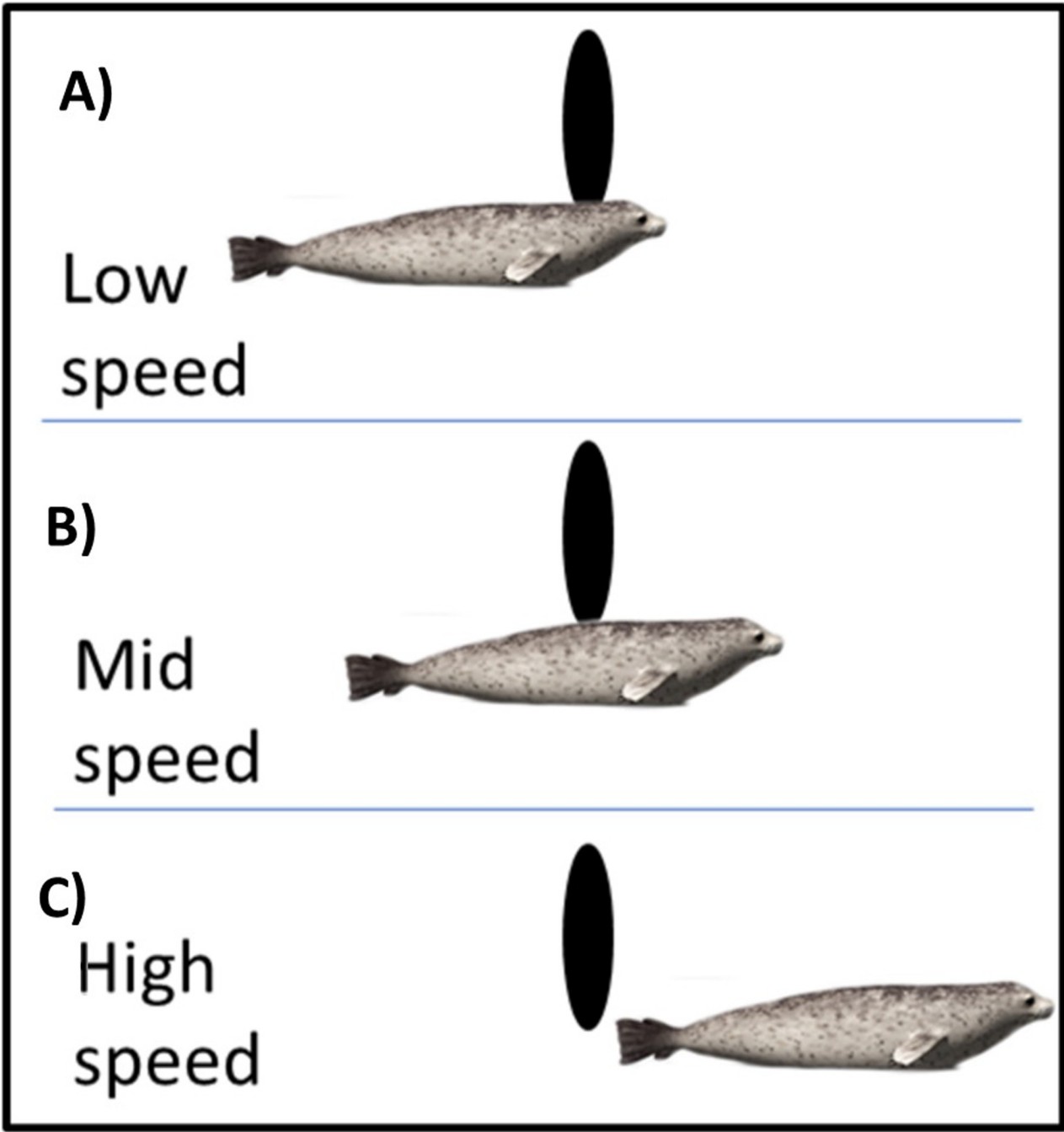

**Fig 5. Diagram displaying three potential interactions between a rotating blade (black ellipsoid) and a seal that are likely to occur due to changes in animal approach speed in relation to the speed of the blade.** A) Displays a common occurrence with low approach speed, where the animal is moving so slowly that most collisions occur with the head. B) Demonstrates how at mid speed the seal is often moving fast enough for the head to pass the rotating blade before a collision occurs with the torso. C) Represents how at a high speed the seal is often beyond the swept area of the device before the blade passes.

animal had a faster (1.8ms$^{-1}$) approach speed, the proportion of head collisions are reduced because the head of the animal had a greater chance of passing the rotating blade more often, resulting in more collisions with the torso (Fig 5B). However, the fastest approach speed (4.0ms$^{-1}$) led to a reduced number of collisions near the tip of the blade as the animal was

moving faster than one full rotation of the blade and so it would no longer collide (Fig 5). This produced more collisions near the hub (Fig 1C), where the collision speeds for the 4.0ms$^{-1}$ scenario showed a peak near 4.0ms$^{-1}$ indicating a higher proportion of collisions near the hub, which will only occur with the head of the seal. However, it should be noted, that these changes to proportion as approach speed increases happen whilst the overall number of collisions reduces.

We demonstrated how one could take a precautionary approach by determining all head collisions irrespective of speed as fatal, assuming concussion and drowning, however a different speed threshold to those applied to torso collisions could also be used as a less conservative approach. The potential for animals to perform some evasive action (e.g., evading a direct strike from a blade) is not considered in existing CRMs; rather, it is typically considered to be a proportion of the number of collisions calculated, as agreed with the regulator. Based on the scenarios used here, a large proportion of collisions often occur close to the hub of the device at low collision speeds (Fig 1). If evasive action is presumed in these scenarios, the animal could evade collision entirely, or collision could occur with the torso, rather than the head and, given that the collision is close to the hub, and the relative speed will be lower as compared to the blade tip, the likelihood of the collision to be fatal is reduced. Therefore, the spatio-temporally detailed outputs from the simulation-based model allow for further consideration as to what a realistic scenario regarding evasive behaviours may be. These scenarios could be informed by what we know about seal behaviour at sea, based on telemetry tag data, for example [15, 28]. This offers one method for further refining the estimates based on the best available science and expert opinion, until suitable empirical data on fine-scale animal behaviour around TECs are available for inclusion in CRMs.

Hypothetical collision speed thresholds that were loosely based on empirical investigations of severe trauma [20] were used to demonstrate how the simulation-based approach can be used to provide probabilities of collision and mortality over a range of speeds. This is a significant development to the traditional CRM approaches, which typically produce a single output value (i.e., number of animals colliding), over one or very few scenarios [13, 17]. Furthermore, the 2D nature of those approaches may underestimate collision speeds as they consider all collisions to occur perfectly perpendicular (i.e., a blade moving perpendicular to a seal) and use only the speed of the blade in the calculations [21]. However, the blade and seal are 3D bodies that are moving, therefore when colliding they create a resultant force that is a combination of their two velocities, which has been addressed herein when calculating the speed of collision.

The ability to consider multiple scenarios based on empirical evidence or expert opinion, using a flexible and robust modelling approach that can incorporate uncertainty, offers considerable advantages to industry and regulators. This will increase the understanding as to how uncertainty may impact upon the outputs from CRMs, and, in turn the predicted risk to a protected population or management unit. For example, the 95% confidence intervals around collision speed for harbour and grey seal mortality (3.2–6.6ms$^{-1}$) calculated by Onoufriou et al [20], could be used to estimate a degree of uncertainty for the mortality estimate. Such as, using the lower bound of the confidence interval (3.2ms$^{-1}$) as a precautionary value for a vulnerable population, or where there is little information on the population's status, and this could be readily updated as and when more information becomes available. The ability to adapt this approach to consider different scenarios is important, as key parameters are often site and context specific, for example, status of population [29], behaviour of animals (e.g., dive behaviour) [30], importance and use of the area (e.g., feeding ground, haul out sites, breeding colonies) [17] and seasonal and/or diel variations in presence [31].

In conclusion, this paper outlines a novel method that advances collision risk modelling through improved estimates of mortality that considers both the speed and location on the

animal where a collision occurs. This robust, flexible, and comprehensive tool for estimating collision risk and mortality on marine wildlife is therefore a significant improvement on existing methods (REFs) for assessing the impacts of collision risk, which have long been a barrier to build out of TEC arrays [12]. This contribution will assist in better understanding and accounting for uncertainty in collision risk estimates and their predicted outcome with respect to collisions being fatal or not. Consequently, this step-change in the simulation-based approach to collision risk modelling, with the inclusion of speed of collision and the potential to apply thresholds to estimate mortality, will be valuable to several key stakeholders, including the TEC industry, statutory nature conservation bodies, and the regulator. In the future, further research into the fine-scale distribution of marine mammals around TECs [15, 28], may provide the best opportunity to refine estimates by incorporating site specific data in to the model. In the absence of empirical data, it is important that we continue to develop a framework of collision risk modelling that can facilitate a better understanding of the potential risk of collision and mortality by providing information on uncertainties and confidence intervals around these collision risk estimates. This approach, as demonstrated herein, offers a solution to this and can be adapted to incorporate new empirical data, or expert opinion, with ease as and when it becomes available.

## Acknowledgments

The Bryden Centre project is supported by the European Union's INTERREG VA Programme, managed by the Special EU Programmes Body (SEUPB). The views and opinions expressed in this paper do not necessarily reflect those of the European Commission or the Special EU Programmes Body (SEUPB).

## Author Contributions

**Conceptualization:** Nicholas Horne, Ross M. Culloch, Pál Schmitt, Louise T. Kregting.

**Formal analysis:** Nicholas Horne, Ross M. Culloch, Pál Schmitt, Louise T. Kregting.

**Investigation:** Nicholas Horne.

**Methodology:** Nicholas Horne, Ross M. Culloch, Pál Schmitt, Louise T. Kregting.

**Supervision:** Louise T. Kregting.

**Writing – original draft:** Nicholas Horne.

**Writing – review & editing:** Nicholas Horne, Ross M. Culloch, Pál Schmitt, Ben Wilson, Andrew C. Dale, Jonathan D. R. Houghton, Louise T. Kregting.

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
