## [Decision Letter · Decision Letter 0]

12 Jul 2022

PONE-D-22-12846Providing a detailed estimate of mortality using a simulation-based collision risk modelPLOS ONE

Dear Dr. Horne,

Thank you for submitting your manuscript to PLOS ONE. After careful consideration, we feel that it has merit but does not fully meet PLOS ONE’s publication criteria as it currently stands. Therefore, we invite you to submit a revised version of the manuscript that addresses the points raised during the review process.

Please try to improve your manuscript and respond to all the reviewers' comments.

We look forward to receiving your revised manuscript.

Kind regards,

Quan Yuan, Ph.D.

Academic Editor

PLOS ONE

Journal Requirements:

Reviewers' comments:

Reviewer's Responses to Questions

**Comments to the Author**

1. Is the manuscript technically sound, and do the data support the conclusions?

Reviewer #1: Yes

Reviewer #2: Partly

2. Has the statistical analysis been performed appropriately and rigorously? 

Reviewer #1: Yes

Reviewer #2: No

3. Have the authors made all data underlying the findings in their manuscript fully available?

Reviewer #1: Yes

Reviewer #2: No

4. Is the manuscript presented in an intelligible fashion and written in standard English?

Reviewer #1: Yes

Reviewer #2: Yes

5. Review Comments to the Author

Reviewer #1: It is an interesting study. A few comments are presented as follows:

1. The contributions of this study are suggested to add.

2. How to guarantee the accuracy of simulation?

3. The writing needs more work.

Reviewer #2: This study estimated the mortality of collision risk between a device and animal using a simulation-based collision risk model. Overall, this study is well organized and can be easily understood. However, this study did not clearly demonstrated the significant contribution in this area. A number of studies have been conducted in this area. This paper can be further improved by more clearly demonstrating the advances in analytical methods, data sources and identification of new significant variables, over the existing studies. What are the new and different findings compared with those from previous studies, and why are these different finding more reasonable?

6. PLOS authors have the option to publish the peer review history of their article (what does this mean?). If published, this will include your full peer review and any attached files.

Reviewer #1: No

Reviewer #2: No

---

## [Author Response · Author response to Decision Letter 0]

22 Aug 2022

Dear Editor,

Please find below our responses to the reviewer comments. We have outlined our replies to each of the points raised by the reviewers in blue and referenced the lines within the manuscript where changes have been made. 

We found the review useful in enhancing the manuscript and thank the editor and reviewers for considering our paper and aiding in its improvement. 

Yours faithfully,

Nicholas Horne

Responses to Reviewer's Questions

Comments to the Author

1. Is the manuscript technically sound, and do the data support the conclusions?

Reviewer #1: Yes

Reviewer #2: Partly

Reviewer 2 provides further information on their response, below. We have addressed them in detail, below..

2. Has the statistical analysis been performed appropriately and rigorously? 

Reviewer #1: Yes

Reviewer #2: No

With respect to Reviewer 2, the paper does not incorporate a hypothesis driven statistical analysis, rather the analysis of data is explanatory in nature, we expect this is why reviewer 2 has responded no here. We note that Reviewer 2, below, asks that we clearly demonstrate the advances in analytical methods, which we have done. We provide more information on this below in response to their specific question. .

3. Have the authors made all data underlying the findings in their manuscript fully available?

Reviewer #1: Yes

Reviewer #2: No

The data has been uploaded to a github repository and access provided through the resubmission process.

4. Is the manuscript presented in an intelligible fashion and written in standard English?

Reviewer #1: Yes

Reviewer #2: Yes

5. Review Comments to the Author

Reviewer #1 

It is an interesting study. A few comments are presented as follows:

1. The contributions of this study are suggested to add.

Descriptions on the significance of the contributions this study has made have been made clearer and expanded upon in the abstract, introduction and discussion. We have now included how this method is novel and is a step-change beyond previous iterations of this model, and how this improvement will benefit key stakeholders (e.g., TEC industry, statutory nature conservation bodes, and the regulator). Please see lines 23, 33, 35-36, 101-114, 316-321, 283-298, 367-368, 371, 383-395, 400-401.

2. How to guarantee the accuracy of simulation?

This model has been published previously with explanations on the accuracy of the simulations. Reference to this peer-reviewed work has now been included in the text of this manuscript. (Lines 127-128)

3. The writing needs more work.

The grammar and spelling have been checked carefully throughout the manuscript and any superfluous text has been removed. 

Reviewer 2 comments

Reviewer #2: This study estimated the mortality of collision risk between a device and animal using a simulation-based collision risk model. Overall, this study is well organized and can be easily understood. However, this study did not clearly demonstrated the significant contribution in this area. 

The significant contributions and novelty of this work have been expanded upon in the text throughout the manuscript and we have now included a summary section in the discussion highlighting the contribution. Please see lines 23, 33, 35-36, 101-114, 316-321, 283-298, 367-368, 371, 383-395, 400-401.

A number of studies have been conducted in this area. This paper can be further improved by more clearly demonstrating the advances in analytical methods, data sources and identification of new significant variables, over the existing studies. 

We have made edits in the text to outline how the methods in this manuscript are different and an advancement on other methods (which also addresses, in part, the comment above from Reviewer 1) and an emphasis on the importance of the ability to now extract more detailed information on what we have now demonstrated are significantly important variables when predicting collision risk (i.e. speed of collision and location of the collision on the body of the animal). Please see lines 32, 287-290, 437.

What are the new and different findings compared with those from previous studies, and why are these different finding more reasonable?

As per Reviewers 1 comments, descriptions on the significance of the contributions this study have been made clearer and expanded upon in the abstract, introduction and discussion (Lines 23, 33, 35-36, 101-114, 316-321, 283-298, 367-368, 371, 383-395, 400-401). We have now included how this method is different and is a step-change beyond previous iterations of this model, and how this improvement will benefit key stakeholders (e.g., TEC industry, statutory nature conservation bodes, and the regulator). Please see lines 330-332, 383-395, 400-401.

---

## [Decision Letter · Decision Letter 1]

13 Oct 2022

Providing a detailed estimate of mortality using a simulation-based collision risk model

PONE-D-22-12846R1

Dear Dr. Horne,

We’re pleased to inform you that your manuscript has been judged scientifically suitable for publication and will be formally accepted for publication once it meets all outstanding technical requirements.

Kind regards,

Quan Yuan, Ph.D.

Academic Editor

PLOS ONE

Additional Editor Comments (optional):

Reviewers' comments:

Reviewer's Responses to Questions

**Comments to the Author**

1. If the authors have adequately addressed your comments raised in a previous round of review and you feel that this manuscript is now acceptable for publication, you may indicate that here to bypass the “Comments to the Author” section, enter your conflict of interest statement in the “Confidential to Editor” section, and submit your "Accept" recommendation.

Reviewer #1: All comments have been addressed

Reviewer #2: All comments have been addressed

2. Is the manuscript technically sound, and do the data support the conclusions?

Reviewer #1: Yes

Reviewer #2: Partly

3. Has the statistical analysis been performed appropriately and rigorously? 

Reviewer #1: Yes

Reviewer #2: Yes

4. Have the authors made all data underlying the findings in their manuscript fully available?

Reviewer #1: Yes

Reviewer #2: Yes

5. Is the manuscript presented in an intelligible fashion and written in standard English?

Reviewer #1: Yes

Reviewer #2: Yes

6. Review Comments to the Author

Reviewer #1: All the reviewer comments have been addressed. Therefore, the reviewer recommends the paper for publication.

Reviewer #2: (No Response)

7. PLOS authors have the option to publish the peer review history of their article (what does this mean?). If published, this will include your full peer review and any attached files.

Reviewer #1: No

Reviewer #2: No
